# Comparison of Different Cure Monitoring Techniques

**DOI:** 10.3390/s22197301

**Published:** 2022-09-26

**Authors:** Alexander Kyriazis, Christian Pommer, David Lohuis, Korbinian Rager, Andreas Dietzel, Michael Sinapius

**Affiliations:** 1Institut für Mechanik und Adaptronik, Technische Universität Braunschweig, 38106 Braunschweig, Germany; 2Institut für Mikrotechnik, Technische Universität Braunschweig, 38124 Braunschweig, Germany

**Keywords:** cure monitoring, fibre-reinforced composite, sensor integration, film sensor, piezoelectric, epoxy, refractive index, impedance spectroscopy, permittivity, dielectric

## Abstract

The ability to measure the degree of cure of epoxy resins is an important prerequisite for making manufacturing processes for fibre-reinforced plastics controllable. Since a number of physical properties change during the curing reaction of epoxy resins, a wide variety of measurement methods exist. In this article, different methods for cure monitoring of epoxy resins are applied to a room-temperature curing epoxy resin and then directly compared. The methods investigated include a structure-borne sound acoustic, a dielectric, an optical and a strain-based observation method, which for the first time are measured simultaneously on one and the same resin sample. In addition, the degree of cure is determined using a kinetic resin model based on temperature measurement data. The comparison shows that the methods have considerable but well-explainable differences in their sensitivity, interference immunity and repeatability. Some measurement methods are only sensitive before and around the gel point, while the strain-based measurement method only reacts to the curing from the gel point onwards. These differences have to be taken into account when implementing a cure monitoring system. For this reason, a multi-sensor node is suitable for component-integrated curing monitoring, measuring several physical properties of the epoxy resin simultaneously.

## 1. Introduction

Composite structures excel in structural applications where a high stiffness-to-weight ratio is beneficial. This is especially apparent in the aerospace industry where the use of composite structures has increased significantly [1]. While the stiffness-to-weight ratio is superior to most materials, there are a number of disadvantages, namely the anisotropy, recyclability and manufacturability. Cure monitoring mainly concerns manufacturability. One big challenge in manufacturing is the exact timing of the curing reaction. Incomplete curing results mainly in a weaker thermo-mechanical load capacity [2,3]. Usually, this creates an incentive to over-cure the material with higher safety values to avoid the problem of undercuring. The solution to unnecessary higher safety margins is the use of in-line cure monitoring systems. They can also provide information for the implementation of smart cure cycles, which allow to reduce residual stresses by a temperature profile adapted to the actual degree of cure [4,5]. Additionally, in-line sensing methods allow better traceability of defects detected later down the production line.

While there are a number of solutions for the monitoring and detection of cure, such as the classical differential calorimetry [6,7,8], DSC for short, or in vitro dielectric measurement [9], many of the methods do not offer the option to measure continuously and directly in the production line.

The main methods for in-line cure monitoring can be subdivided in three main working principles: optical, electrical and acoustic. Two main points of view can be distinguished in optical methods. One is the direct optical measurement of reaction properties by means of spectral absorption of the main reaction components. These properties are utilised by the Raman spectroscopy [10,11,12,13,14,15,16,17,18,19,20,21]. The other methods exploit the change in the refractive index and are rather unusual [22,23,24,25,26,27,28,29,30,31,32,33,34,35]. In this article, the changes of the refractive index are measured by means of an optical prism. This is a novel approach, but not applicable for practical measurements in fibre-reinforced parts. It therefore serves mainly for comparison of the measured parameters.

The second way to measure the current state of cure is the utilisation of the change in electrical, usually dielectric, properties of the curing epoxy resin. As the resin curing proceeds, the electrical behaviour changes from an ion-conducting liquid to a dielectric solid. There are three main physical effects occurring in curing epoxy resin, namely, ion conduction, electrode polarisation and dipole relaxation. The different physical effects cause the electrical properties to be dependent of the time, degree of cure and temperature. All this information can be reconstructed from the microscopic electrical impedance of the resin. The different cure monitoring systems can be distinguished by their data reduction and are thus more sensitive to several of the physical effects. Most monitoring techniques use sensors with interdigitated electrodes, which form a capacitor when exposed to a dielectric, non-conducting medium [36,37,38,39,40,41,42,43,44,45,46,47,48,49,50,51,52,53,54] but also plate capacitors [55,56,57,58] or more exotic techniques are used [59,60,61]. Some authors examine the real and imaginary parts of the complex impedance of the interdigitated electrodes or magnitude and phase of the impedance [45,50,51,54,57,62]. The electric loss factor has been used as a measure for the curing as well [38,39,40,42,43]. Sometimes only a single frequency is taken into account, allowing simpler electronics for evaluation [38,39,40,44,57] but often a whole spectrum is considered, often in the range between 1 Hz and 1 MHz [36,37,41,45,46,55,56]. The usage of DC excitation [58] or step response analysis [47,48] is more exotic. As another macroscopic measure, the capacitance [47,59] and resistance [44,47,58,61] of the interdigital sensor can be calculated from impedance data. Using modelling of the electrical field between the sensor electrodes, microscopic measures such as ion viscosity or permittivity can also be calculated [36,37,41,45,46,48,49,53,56,63,64,65,66]. Often, correlations to the degree of cure obtained from DSC measurements [36,37,38,39,40,42,43,45,57,59] as well as to the viscosity [36,37,41,57,58,59] are identified.

The third main angle to tackle cure monitoring is acoustic measurement. Acoustic cure monitoring concentrates on the change in density and stiffness over time. Classical methods focus on pulsed through material measurements [67,68,69,70,71,72,73]. Newer approaches consider the tool and the curing composite as one conglomerate medium for lamb wave propagation [74,75], while another relatively new technique developed by Pommer et al. utilizes wave reflections at the boundary between tool and material to gain information on the curing [76,77].

This paper aims to compare the three main vantage points for in-line cure monitoring with each other by means of simultaneous measurement. The measurements are carried out simultaneously on the same resin sample for two reasons: Firstly, the curing reaction of epoxy resins is sensitive to mixing deviations. With simultaneous measurements on the same resin sample, mixture deviations affect all measurements in the same way. On the other hand, simultaneous measurement provides information about possible interactions between the measurement methods. This is important if several measurement principles are used together in a later application for the purpose of data fusion. The investigation of strain development during epoxy curing includes one of the more rarely used cure-monitoring techniques [78], which is also sometimes performed using optical fibres [79]. The comparison of multiple measurement techniques in the same specimen is new in the extent presented in this article. A kinetic simulation model serves to establish the relation of the four techniques to the degree of cure.

## 2. Materials and Methods

### 2.1. Overview of the Complete Measurement Setup

All experiments are conducted for the resin system RIMR 426 with curing agent RIMH 435 (Lange + Ritter, Germany). The resin system is a room-temperature curing epoxy resin whose reaction rate is mainly sensitive to temperature; therefore, temperature is one of the measured parameters during the experiments. The complete measurement setup is shown in Figure 1. The peculiarities of the parts of the experiment are explained in the following subsections. The full setup consists of a strain gauge, thermocouple, an acoustic measurement based on reflected ultrasonic waves, and an interdigitated sensor for impedance measurement. An Arduino serves to collect the measured acoustic spectra and the temperature measurements. It also reads out an HX711 connected via a wheatstone bridge to the strain gauge. From the Arduino, the measured data are transferred to a Raspberry Pi, which adds a camera image for the analysis of the change in refractive index to the data set and saves the data. Since the electrical impedance measurements require a personal computer with a special software, these measurements are stored there. The synchronisation is ensured by a clock comparison between both computers and the simultaneous start of the measurements. The aluminium plate with the piezo ceramics is placed inside the mould during the experiments as indicated by the dashed lines. In Figure 1, it has been removed from the mould to make the sensors visible, which are integrated into the resin.

### 2.2. Refractive Measurement

The refractive measurement is based on the change in the refractive index during curing. To magnify the changes in the refractive index, a prism is set up with a 3D-printed backbone, and microscope glass panes are glued into the prism to form the prism angle of 60°. A standard red laser pointer is used to create a light beam that can be detected on a screen as depicted in Figure 2. The prism is aligned towards the laser pointer so that the light beam strikes the surface under an angle of 35°. The screen is oriented towards the prism so that the light beam strikes it almost orthogonally during the movement of the light beam during curing. Using a camera behind the screen, the movement of the light beam during epoxy curing can be recorded by taking pictures of the screen every two minutes and evaluating the light dot position by image processing each photo.

The centre of the laser beam is calculated for each image. Images with red, green and blue channels, each represented by 8-bit integers, are used as input for the algorithm. Since the laser beam is monochromatic red light, the algorithm starts by extracting only the red channel. This channel is then binarised by applying a threshold value. All pixel values above this value appear white and all others appear black in the resulting image. To investigate the repeatability, four experiments were conducted. The exact threshold values for each experiment are listed in Table 1.

The reason why the images were not analysed in the second experiment is explained further down in the article. The threshold is lower in the first experiment than in the third and fourth experiments. Due to strong scattering of the laser beam, the brightness is lower than usual. In order to obtain as many data points as possible, a lower threshold is necessary. Even with this modification, not every centre point can be computed, which leads to missing data points in the first experiment, see Figure 8 in the results section. The binarised image is further processed by extracting the coordinates of the white pixels. The coordinates $x_{c}$ and $y_{c}$ can then be used to compute the centre points in the x and y directions. For that, the following equations are used, respectively, where $x_{w}$ and $y_{w}$ denote the coordinates of a single white pixel and *W* denotes the number of white pixels.
(1)$$x_{c}=\frac{\sum x_{w}}{W}\;y_{c}=\frac{\sum y_{w}}{W}$$

In addition to the computation of the centre point, the algorithm also extracts the timestamp and converts the coordinate unit from pixel to millimetre. Exif tags are used to extract the timestamp. In order to convert the units, a conversion constant is needed. This constant is also extracted automatically using a green sticker with known size in the shape of an 8 mm diameter circle placed in the camera view for calibration. This circle always appears black in the red channel and therefore can be extracted by binarising the image with a threshold of 30. Each pixel below this value is part of the circle and appears white in the binarised image. Since there are always objects at the edge of the image, which also appear white, a stripe of 70 pixels is cut off on each side. The size is then reduced from 640 × 480 to 500 × 340 pixels. The difference in the maximum and minimum coordinates in the *x* and *y* directions is computed, added and divided by two in order to obtain an estimation for the circle diameter in pixels from which the conversion constant is computed.

From the propagation of the light beam through the prism, the relation between the light dot movement on the screen and the change in the curing resins refractive index is derived. According to the technical data sheet, the epoxy resin RIMR 426 has a refractive index of 1.553 to 1.556, and the curing agent RIMH 435 has a refractive index of 1.483 to 1.486. A refractive index of $n_{0}=1.55$ is taken for the simulation. Since light dot movements *x* between 0 and 50 mm can be observed in the experiments, the data from the ray tracing algorithm are condensed into a polynomial that is fitted to the data. The *m* in the formula denotes the SI base unit meter.
(2)$$\frac{\Delta n}{n_{0}}=-2.0569\;\cdot \;{\left(\frac{x}{m}\right)}^{2}+0.4761\;\cdot \;\left(\frac{x}{m}\right)-1.7546\;\cdot \;10^{-6}$$

The polynomial fit describes the relation between the light dot movement and the relative change in the refractive index well, as shown in Figure 3. The experimental section shows the changes in refractive index calculated from the light dot movement using the fitted polynomial.

### 2.3. Ultrasound Resonance Spectroscopy

In ultrasound resonant spectroscopy, the tool—in our studies, a thin aluminium plate—is used to guide ultrasound waves to the interface between the resin and tool. The transmission and reflection of acoustic emissions at different excitation frequencies create an acoustic spectrum depending on the acoustic impedance of boundary conditions and tool properties. While the acoustic properties of the tool do not change in a noticeable way during the measurement, the properties of the epoxy resin change to a considerable extent. This creates an absorption effect at the boundary conditions of up to 30% of the incoming signal for an aluminium–epoxy pairing. Detailed information can be found in [76,77,80].

The particular setup for this method consists of two 15 mm piezo actuators (PI Ceramic GmbH, Germany), which are glued to a 3 mm aluminium plate by a standard cyanoacrylate adhesive. The plate serves equally as a tool and electric grounding for both sensors. This way, electric noise between both piezo actuators should be minimal. One major drawback is the possibility of interference and hum loops with other equipment where the aluminium plate must also be grounded, namely the dielectric measurement.

The operating frequency range is 100 kHz to 1 MHz. This is lower than the resonance frequency of the plate in the thickness direction, but this is usually not a problem due to the frequency-independent absorption. However, it reduces the accuracy due to a lower signal-to-noise ratio.

The measurement system uses a band-pass filter between 100 kHz to 1 MHz. The signal acquired is assumed to be only or at least mostly the reflected signal at excitation frequency. The filtered signal is then processed by a true RMS-DC component at multiple excitation frequencies, resulting in a transfer function for the excited frequencies.

The change of the mean value of all excited frequencies is a measure for the current degree of cure.

### 2.4. Dielectric Measurement

For the dielectric measurement, a dielectric sensor with interdigitated electrodes is used. The sensor design and manufacturing as part of a multiple sensor node were described in [81]. Both electrodes consist of 15 fingers each with 100 μm width, 3.75 mm length and 30 μm distance between them. The sensors have a capacitance in air of 3.8 pF. For monitoring the permittivity changes during curing, the electrical impedance of the sensor is measured using a four-wire-circuit by a Gamry Reference R600+ potentiostat (Gamry Instruments Inc., Warminster, PA, USA) in the frequency range from 3 Hz to 100 kHz. Electrical spectra are recorded every 100 s, which is near the maximum operating speed of the potentiostat. To reduce noise sensitivity, the ground electrode of the potentiostat is connected to the aluminium plate for the second, third and fourth experiments. In the first experiment, the ground electrode is not connected. The aluminium plate also serves as ground for the piezoceramics in the ultrasound resonance spectroscopy system.

### 2.5. Strain and Temperature Measurement

Figure 4 depicts the basic principle of the strain gauge measurement for determining the degree of cure. A type 6/350LC11 strain gauge (350 Ω, HBM, Germany) is attached by adhesive bonding over a notch at the bottom of the prismatic mould. Since the epoxy resin experiences a volume reduction during cure, the polymer volume between the strain gauge and the mould also shrinks. Up to the gelation point, the shrinkage can be counterbalanced by liquid flowing from the upper resin volume, reducing the liquid level of the prism. When the epoxy resin has passed the gel point, no further resin flow is possible from the upper volume, so the strain gauge is deformed as depicted, with high exaggeration, in Figure 4, and thus it experiences a stretching due to the clamping on both sides of the notch in the mould. This measurement setup is not particularly suitable for an application, but it is sufficient to show the feasibility of the measurement principle.

The strain gauge is connected to a wheatstone bridge that was calibrated before the measurement. A standard HX711 chip is used for powering the bridge and reading out the changes in the bridge voltage. The HX711 expresses the bridge voltage referred to the supply voltage as a numeral, *N*, which can be converted into the relative resistance change using the following formula:(3)$$\frac{\Delta R}{R}=\frac{4\;\cdot \;N}{2^{23}}\;\cdot \;\frac{20mV}{5V}=1.907\;\cdot \;10^{-9}\;\cdot \;N$$

The numeral *N* is a signed 24 bit integer, resulting in the denominator $2^{23}$. The bridge supply voltage is 5V, the maximum bridge voltage is ±20mV for the HX711, and factor 4 indicates that a single strain gauge is used in a wheatstone quarter bridge. A standard type K thermocouple is used for the temperature measurement in combination with a MAX31850 chip, which is connected to the Arduino microcontroller for data acquisition as well as the HX711.

### 2.6. Kinetic Cure Model

A kinetic reaction model based on DSC measurements is used to verify the different measurement methods. This model uses temperature information obtained by a discrete type K temperature probe embedded in the epoxy to approximate the degree of cure. The model is obtained by the fitting of previously acquired high accuracy differential scanning calorimetry measurements.

The fit is performed with the following kinetic base model equation: (4)$$\begin{array}{cc} \frac{d\alpha }{dt} & =k_{1}{\left(\alpha_{max}-\alpha \right)}^{l}+k_{2}\alpha^{m}{\left(\alpha_{max}-\alpha \right)}^{n}\\ k_{i} & =A_{i}e^{\left(E_{i}\;/\;RT\right)}\;i=1,2 \end{array}$$(5)$$\begin{array}{cc} \alpha_{max} & =2.538\;\cdot \;10^{-7}T^{3}-2.941\;\cdot \;10^{-4}T^{2} \end{array}$$(6)$$\begin{array}{cc} \; & +1.144\;\cdot \;10^{-1}T-13.89 \end{array}$$
where *T* is the temperature in K, *R* is the universal gas constant, *t* is time, α is the degree of cure, and $\alpha_{max}$ is the maximum achievable degree of cure for the current temperature. Table 2 shows the best fits for the model parameters *k*, $A_{i}$, $E_{i}$, *l*, *n* and *m* using the CoPE tool [82]. A comparison between the DSC measured curves and the fitted model is shown in the appendix in Figure A1.

The Flory polymer theory yields the gelation point of the epoxy resin at approximately 57.7% of the maximum achievable curing value [83].
(7)$$\alpha_{Gel}=\frac{1}{\sqrt{\left(f_{resin}-1\right)\;\cdot \;\left(f_{agent}-1\right)}}=\frac{1}{\sqrt{3}}$$
where *f* is the number of epoxy groups of the resin $f_{resin}=2$ and amine hydrogens of the curing agent $f_{agent}=4$, respectively. For reference, the resin and the hardener are composed of the components described in Table 3. All resin components consist of molecules with two epoxy groups, and all curing agent components consist of molecules with two primary amino groups, i.e., four amino hydrogen atoms.

### 2.7. Rheology Measurements

Samples of the same resin system are also measured for gel point determination in a rheometer (Anton Paar, Austria) during cross-linking at the cross-linking temperatures 40 ${}^{\circ }C$, 60 ${}^{\circ }C$ and 80 ${}^{\circ }C$. In order to take into account the progressive cross-linking of the resin, oscillating shear deformations are used, which, starting from 10%, are reduced time controlled to shear deformations of 0.1%. Parallel plate configuration is used with a disc diameter of 25 mm and a distance of 1 mm.

## 3. Results

The type K thermocouple displayed in Figure 1 produces the temperature data shown in Figure 5. The figure shows that heat is produced by the exothermic reaction of the resin during the first hours. The ambient temperature was around 3 K lower during the first experiment.

Using this temperature data and the kinetic Equation (Equation 4), the degree of cure over time shown in Figure 6 is obtained. The degree of cure does not reach 100% but converges asymptotically to its maximum value of 78.5% for a temperature around 26 ${}^{\circ }C$ in the end of the experiment. According to the Flory theory, the gelation times are found at 57.7% degree of cure. Since the ambient temperature during the first measurement is around 3 K lower than during the second, third and fourth measurements, the gelation point is reached considerably later in the first experiment. The model predicts a difference of around 2 h in gel time. This highlights the importance of cure monitoring since small temperature deviations can cause remarkable changes in the gel time. The calculated gelation points are also included in all following figures for comparison.

The rheological investigations show a good agreement between the gel point from the Flory theory and the degree of cure, where storage and loss moduli are equal; see Figure 7. While the Flory theory predicts the gel point at 57.7% for the resin system used, in the rheometer test, the gel points can be found at about 60% (40 ${}^{\circ }C$), 61% (60 ${}^{\circ }C$) and 57% (80 ${}^{\circ }C$). The curves in Figure 7 show that the rheometer is no longer able to measure the modulus of the epoxy resin beyond 1 MPa.

A number of interferences between the different measurement methods as well as general unpredictable disturbances occurred during the experiments. Firstly, in the second experiment, air bubbles formed in the optical path during curing, creating additional refractional boundaries and blurring the laser image on the screen. Secondly, poor grounding in the first experiment caused errors in the dielectric measurement and made the data unusable. Thirdly, interference between the dielectric and the acoustic measurement occurred as the electronics used to measure the acoustic signal were not adequately protected against electrical interference. This resulted in partially or fully unusable measurement data when both measurements were active at the same time. Table 4 lists the successful, partial successful and failed measurement attempts during the individual experiments.

The relative change in refractive index calculated by the image processing algorithm using the correlation with the light dot movements (Figure 3) is shown in Figure 8. The data show that the curing reaction causes visible changes in the refractive index up to 14 h after mixing (third and fourth experiment) and 16 h for the first experiment. Due to the binarisation algorithm, the red light dot could not be detected in all images during the first experiment, as described above in the materials and methods section. This is visible in Figure 8 as gaps in the curve.

The experiments on curing monitoring using resonance ultrasound spectroscopy show unexpected results, with the exception of experiment 1 (Figure 9), due to interference between the different electrical measurement methods. For commercial curing process monitoring, these results would be considered unsatisfactory, but from a scientific point of view, the results provide insights into the limitations and interferences of the acoustic measurement principle. Firstly, a high noise level is evident in all measurements, except the first experiment. On the other hand, changes in the acoustically reflected amplitude due to the curing reaction could be observed up to 10 h after mixing in the first trial.

In addition to the noise in the data, the repeatability of the acoustic measurements is also very low. This becomes clear when comparing the second experiment with the fourth experiment. On the one hand, the second experiment shows no further change in reflection amplitude already about six hours after mixing. On the other hand, the changes in the initial amplitude of the fourth experiment are visible up to about 12 hours after mixing. There are probably two reasons for the variations between the experiments: Firstly, the embedding situation of the aluminium sheet is decisive for the reflection behaviour of the ultrasonic waves. The use of a new prism for each experiment could therefore have affected the repeatability. Secondly, the dielectric measurement system in the first experiment had no connection to the aluminium plate for grounding. This made the dielectric measurements unusable, but the acoustic measurements were flawless. This indicates that the dielectric and acoustic measurement systems interfered via the common ground electrode and should not work together in spatial and temporal proximity, or should be electrically separated by a galvanic isolation. They could be used with some distance between them or in an alternating mode. In addition, the acoustic measurement system seems to be better suited for tool-integrated cure monitoring, as this form of integration guarantees a higher constancy of the embedding conditions.

The third cure monitoring method explored in this paper is the dielectric measurement by means of an embedded interdigitated electrodes sensor. Since the impedance is measured over a spectrum between 3 Hz and 100 kHz and during the curing process, multiple spectra are recorded, the impedance data can be depicted over the plane spanned by the curing time and the frequency as shown in Figure 10, where the upper part shows the magnitude of the electrical impedance and the lower part shows the phase angle.

The upper part of Figure 10 shows the absolute value of the measured electrical impedance from the second experiment as a surface over the time frequency plane. The data for the third and fourth experiment look similar. The data for the first experiment could not be analysed due to grounding errors. The frequency and impedance axis are scaled logarithmically and for most of the data, the impedance appears inversely proportional to the frequency. This inverse proportionality is typical for a capacitor. For low curing time, the epoxy resin behaves like an ion-conducting liquid, resulting in a ohmic conductor in parallel to the capacitor. Thus the frequency response of the interdigitated electrodes capacitor appears more like a first order low pass filter. With increasing curing time, the cut-off frequency of the low-pass filter shifts to lower frequencies due to the decreasing mobility of ions and the higher viscosity of the epoxy resin. At the same time, the maximum impedance level increases. In the low frequency range, disturbances can be clearly seen. These disturbances belong to the 50 Hz supply frequency and its multiples and also to a frequency band around 20 Hz, whose origin is not yet identified. The lower part of Figure 10 presents the phase angle of the impedance. For a better interpretability, the biggest disturbances at 20, 50 and 100 Hz were removed in this diagram. For a perfect capacitor, the phase angle would be 90° over the whole time–frequency plane. In contrast, the phase angle reaches 0° for low frequencies and low curing times, which corresponds to the low-pass filter behaviour that is also visible in the magnitude of the electrical impedance. The observed behaviour is consistent with the theory that the liquid resin allows ion movement. With the proceeding curing, the viscosity of the resin increases so that the ion movement becomes more and more hindered until the ion movement is no longer detectable after around 5 h of curing.

Another effect is visible after 5 h of curing: At a frequency of 100 kHz a peak can be observed in the phase angle surface, at which the phase angle rises to around −86°. This peak proceeds to lower frequencies with increasing cure time. It belongs to the physical effect of dipole relaxation, which has to be distinguished from the electrode polarisation and ion movement. The effect of dipole relaxation occurs because of dipole molecules that orient in the exciting electric field, yielding a higher relative permittivity at low frequencies. When the dipoles become more and more immobilised with progressed curing, they are no longer able to follow a high-frequency excitation. The effect of dipole relaxation can be modelled as a frequency dependent capacitance using the Cole–Cole model [84].
(8)$$C=C_{\infty }+\frac{C_{0}-C_{\infty }}{1+{\left(j\frac{f}{f_{CC}}\right)}^{\beta }}$$

The relaxed capacitance $C_{0}$ and the high frequency capacitance $C_{\infty }$ only show slight changes during the cure process [85,86] and β only varies between 1 and 0. The most interesting parameter is the dipole relaxation frequency $f_{CC}$ at which the dipoles are barely able to follow the excitation resulting in the loss peak in the phase angle surface. $f_{CC}$ shifts over multiple decades during the process of the curing reaction [55,56,87,88]. At $f_{CC}$, the loss part of *C* reaches a maximum, and the storage part of *C* is exactly $\frac{C_{0}+C_{\infty }}{2}$. When $f_{CC}$ passes low values, this is generally attributed to vitrification of the resin, where the reaction ends or at least slows down gravely. In the literature, the thresholds are often set to 0.1 Hz [55,88,89] or 0.16 Hz [41], which corresponds to $2\pi \;\cdot \;f_{CC}=1$ Hz. Therefore, the measured capacitance decreases, and the measured impedance increases. This effect can hardly be seen in the magnitude of the impedance, but when the capacitance is calculated from the impedance, the effect becomes clearly distinguishable.

For simplification, the dielectric sensor is assumed to be a parallel arrangement of a capacitance and an ohmic resistance. Then, the following equation can be used to calculate the capacitance *C* from the measured complex impedance $Z\left(\omega \right)$:(9)C=Re1jωZ(ω)

The assumption of a parallel arrangement of a capacitance and an ohmic resistance is a dramatic simplification of the real physical conditions. As can be seen in Figure 11, electrode polarisation effects influence the measured capacitance, especially at low frequencies and during the beginning of cure. The effect is not perceptible in Figure 10 and not covered by the simplified parallel arrangement, which has to be kept in mind when interpreting the results.

Figure 11 shows the capacitance calculated from the measured impedance over the time for the frequencies 5 Hz, 125 Hz, 5 kHz and 100 kHz. In the upper two diagrams with low measurement frequencies, the diagrams start with a negative slope, which ends after 5 h for 5 Hz measurement frequency and after 3 h for 125 Hz measurement frequency. These negative slopes belong to the electrode polarisation effects due to ion movement, and mark the time and frequency range in which the interdigitated sensor does not behave like a single capacitor. Despite this effect, all capacitance values start at 13 pF and experience a shift down to 7 to 8 pF later in the curing reaction. That transition belongs to the dipole relaxation effect and occurs earlier for the higher excitation frequencies and later for the lowest excitation frequencies. On the one hand, low frequencies are attractive to measure the progress of the curing as far as possible. On the other hand, the low frequency range exhibits higher disturbances due to interferences with the other measurement techniques. However, the dielectric measurements in all diagrams exhibit very good repeatability. When measuring at 100 kHz, the epoxy resin has experienced most of the transition down to 8 pF after 8 h, when it has just reached the gelation time. If the capacitance is measured at 5 Hz, the epoxy resin is just at the beginning of the transition between 13 pF and 8 pF. However, because of the high disturbances in the curve, the end of the transition cannot clearly be detected. The measurement at 125 Hz appears to be a good compromise between low disturbance and a late transition. In the bottom part of Figure 11, the course of the dipole relaxation frequency $f_{CC}$ from a least squares fit is displayed ($C_{0}$ fixed to 13.0 pF and $C_{\infty }$ fixed to 7.3 pF). Due to high disturbances in the low frequency measurement data, the dipole relaxation frequency can only be tracked down to the range of 100 Hz after 7 h. Vitrification could be observed with a better signal-to-noise ratio, either in a less noisy environment or by an increase in the sensor capacitance in a future redesign.

The final compared in-line cure monitoring technique utilizes the epoxy-shrinkage-induced strain on an integrated strain gauge. The calculated resistance changes of the strain gauge are shown in Figure 12. Since the wheatstone quarter bridge is not temperature compensated, the strain gauge does not only react to the occurring strains but to temperature changes as well. Comparing Figure 12 to Figure 5, the resistance changes between the initial 4 h can be attributed to the temperature development due to the exothermic curing. Comparison of Figure 12 with Figure 6 shows that the stronger changes in the resistance from about 7 h on (for the third and fourth experiments) respectively from 9 h on (first experiment) can be attributed to the reaching of the gelation point and subsequent deformation of the strain gauges by the shrinkage of the curing epoxy. The differences in the onset times of force translation from the epoxy resin to the strain gauge are in good agreement with the differences of the gel times due to varying ambient temperatures between the different experiments.

## 4. Discussion

The test results provide a number of valuable insights into the limits and compatibility of the individual methods for cure monitoring. It is clear from the results of the acoustic experiments that only poor repeatability was achieved in the presented test results. A probable explanation for the poor repeatability are deviations in the insertion process of the aluminium plate or different resin levels, which lead to changes in the reflection behaviour of the structure-borne sound waves. The acoustic measurement method is thus sensitive to minor geometric changes and is therefore less suitable for part integration than for tool integration. However, the acoustic measurement method is suitable for integration in a rigid, closed mould, as used in pultrusion [76,77,80] because the geometry of the mould does not change over time.

In contrast, we found excellent repeatability for the dielectric and optical measurement methods. In the dielectric measurement, the high repeatability is noticeable in the fact that the measured capacitances hardly differ from each other at the end of the curing. Stronger deviations in the time window between 6 and 8 h, on the other hand, can be attributed to the temperature-related different reaction rates in individual measurements. In the case of the dielectric measurement method, the high repeatability is due, on the one hand, to the well-controlled process of sensor manufacture. On the other hand, the local measurement principle, which only measures in the direct vicinity of the interdigitated electrodes regardless of the size of the resin volume, plays a decisive role. In the case of the optical measurement method, high repeatability is achieved, on one hand, by the precise geometry of the prisms. On the other hand, the repeatability of the measurements also indicates that the optical refractive index strongly correlates with the degree of cure.

Regarding electrical disturbances, the dielectric and acoustic measurement methods in particular revealed weaknesses. The dielectric measurement data of the first experiment were unusable due to interfering electromagnetic waves. In contrast, the dielectric measurement data of measurements 2, 3 and 4 could be recorded by connection of the grounding line to the aluminium plate. The high interference sensitivity is due to the high electrical impedances occurring in the sensor. Especially at low frequencies, extremely low electrical currents flow in the sensor so that even slight electromagnetic interference compared to the useful signal can cause a clearly perceptible noise. The analysis of the entire impedance measurement data shows clear disturbances especially at multiples of 50 Hz, sometimes far greater than the intended signal. However, even in the frequency range below 50 Hz, some interference bands can be identified from sources that have not yet been identified. Conversely, unlike the dielectric measurement data, the acoustic measurement data in experiment 1 show minimal interference. After connecting the ground wire to the aluminium plate on which the piezoceramics are glued, the acoustic measurements 2, 3 and 4 showed clearly recognisable disturbances. This observation suggests that the two measurement systems interfere and interact with each other. Against this background, simultaneous operation of the acoustic measuring system in the direct physical vicinity of the dielectric measuring system does not appear to make sense. A sharply limited frequency filter could provide a relief; a carrier frequency amplifier can also be helpful for a significant increase in the signal-to-noise ratio due to the sharper and self-tuning filtering characteristics [77].

The data also reveal that the measuring principles react sensitively in different sections of the curing reaction. The thermal measurement only shows significant deviations from the ambient temperature in the range of high reaction rates. The measured temperature is a result of the heat released during the reaction, the heat outflow into the environment, the internal convection and the local heat capacity. Therefore, the results of the thermal measurement cannot be easily converted into a degree of cure. However, the strength in the thermal measurement method rather lies in the ability to make a model-based prediction of the degree of cure in combination with a kinetic model from DSC measurements. For this, however, the initial degree of cure has to be known. If only the thermal method is available, the entire curing progress has to be observed.

In Figure 13, measurement parameters and the derived variables are plotted against the degree of cure calculated from the specimens temperature history. The sensitivity of the single measurement systems correlates with the slope of the curve. If the slope is zero, the measurement variable is unaffected by the degree of cure. If the absolute value of the slope is high, there is a high dependence of the measurement variable from the degree of cure. The acoustic measurement seems to be sensitive up to the gelation point and just a little beyond. This is consistent with the fact that especially the loss modulus of the resin changes by several magnitudes up to the gel point, as can be seen in Figure 7. In contrast, the strong changes in the storage modulus in the area around the gel point and during the subsequent vitrification seem to exert little influence on the reflected amplitude. This is possibly due to the fact that the stiffness of the vitrified epoxy resin is also still more than a magnitude smaller than that of the aluminium plate to which the piezo actuators are glued. This suggests that the acoustic measurement principle used here reacts more to changes in the loss modulus than to changes in the storage modulus. The optical measuring system reacts somewhat beyond the gel point to the curing reaction, but reaches its final value before the end of cure. Thus, the optical measurement method is also not suitable for tracking the degree of cure over the entire curing reaction.

The dielectric measuring system also reacts sensitively to the curing reaction after the gelation point. Tracking the dipole relaxation frequency down to 0.1 Hz should allow to monitor even the vitrification of the resin. In the experiments presented here, the disturbances caused by the acoustic measurement system impede the tracking of the dipole relaxation frequency to such low values. While the ionic conductivity was no longer detectable at 125 Hz after three hours in the resin system considered, some authors report that they were able to observe the degree of cure over the entire curing reaction with the ionic conductivity [38,39,40,42]. A likely reason for this is that the ionic conductivity is strongly temperature dependent and changes by several orders of magnitude at higher temperatures. This is often modelled by means of an Arrhenius relation. When curing at room temperature, the influence of the dielectric constant therefore dominates from a much earlier point in time so that the electric losses become vanishingly small compared to the capacitive component. In the experiments presented in this paper, the electric energy loss due to ionic conductivity disappeared below the detection limit within the first four hours of the measurement, which is why the calculated capacitance and not the resistance was chosen for the representation in Figure 11. With high-temperature curing, on the other hand, it is to be expected that the influence of the ionic conductivity dominates for much longer so that the curing reaction could possibly be traced to the end on the basis of the electric loss.

The strain measurement has the interesting property that it only reacts to the curing after the gel point. This connection is not surprising, as the epoxy resin only behaves like a solid from above the gel point and begins to transfer forces to the strain gauge in order to deform it. The addition of a strain gauge to the dielectric measurement therefore seems to make sense, especially for low-temperature curing resins. The integration of both measuring principles on the same foil sensor would also reduce the effort required for embedding into fibre composites.

## 5. Conclusions

The article presents simultaneous measurements in a curing epoxy resin volume using different measuring techniques. The results from a kinetic curing model supplied with thermal data, an interdigitated electrodes sensor, a structure-bourne acoustic measurement system, a refractive index measurement technique and a strain gauge are compared. While strain gauges, interdigital electrodes and thermocouples can easily be introduced into the manufactured part and the acoustic measurement system can be integrated into the tool, the optical measurement system cannot be applied to practical manufacturing processes. The measurement results reveal that the interdigitated electrodes sensor and the piezoceramics-based structure-bourne sound system interfere with each other. Thus, electrical grounding is important for both techniques.

Comparison reveals that the different measurement techniques detect the curing reaction during different phases of the curing reaction. On the one hand, the acoustic structure-bourne sound technique and the interdigitated electrodes can detect the curing reaction approximately to the gelation point and slightly beyond. On the other hand, the strain gauge starts to detect the curing reaction when the resin has reached the gelation point. Thermal measurements are not meaningful for the curing reaction without a kinetic curing model. Combining the strain gauge with an interdigitated electrodes sensor can improve the information yield from the curing reaction. The next step is to use the already developed monolithic multisensor node consisting of a temperature sensor, strain gauge and interdigital electrodes [81] for monitoring the curing process of composites.

## Figures and Tables

**Figure 1 sensors-22-07301-f001:**
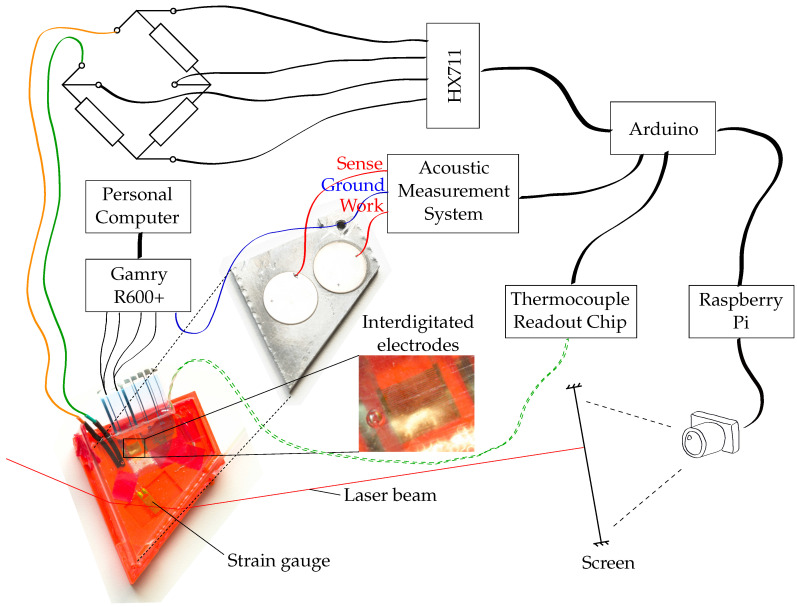
Measurement setup for the experiments described in this article. In the first experiment, the ground wire between the Gamry R600+ potentiostat and the aluminium plate was not connected.

**Figure 2 sensors-22-07301-f002:**
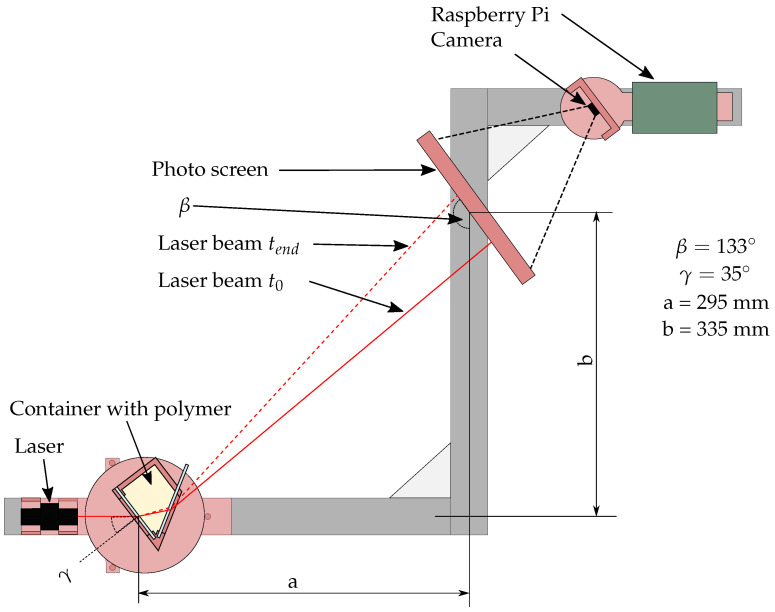
Schematic sketch which illustrates the path of the laser beam through the prism, the arrangement of the photo screen and camera perspective (top view).

**Figure 3 sensors-22-07301-f003:**
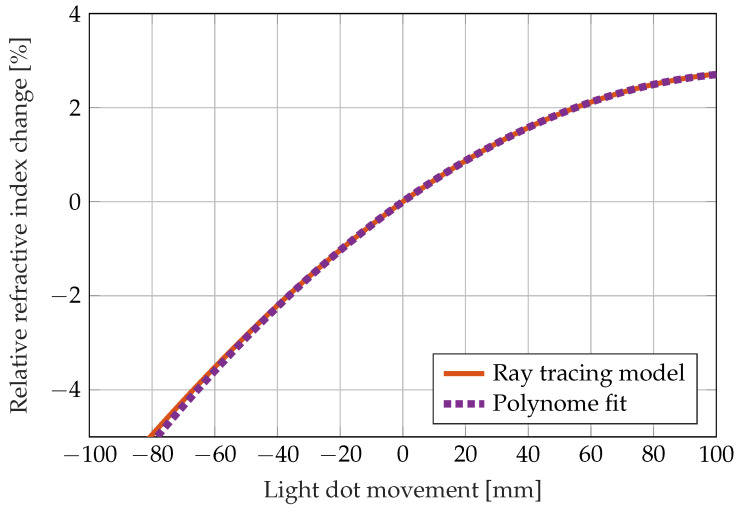
Relation between the visible light dot movement on the screen and the relative change of the refractive index calculated by a ray tracing model and fitted with a polynomial.

**Figure 4 sensors-22-07301-f004:**
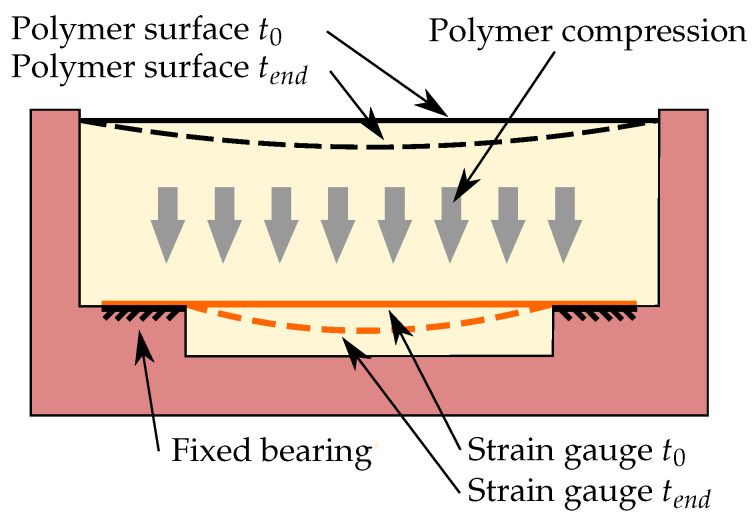
Polymer and strain gauge movement during the experiment. The thickness of the polymer layer is exaggerated to illustrate the effect.

**Figure 5 sensors-22-07301-f005:**
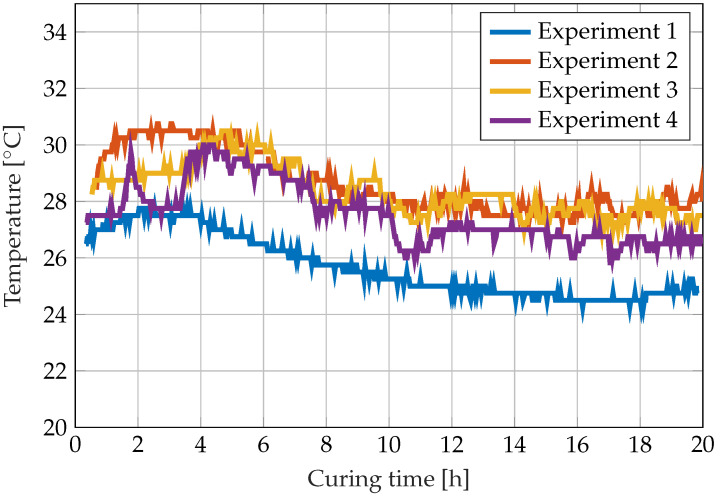
Temperature development during the curing reaction of RIMR 426 resin with RIMH 435 curing agent measured with a type K thermocouple.

**Figure 6 sensors-22-07301-f006:**
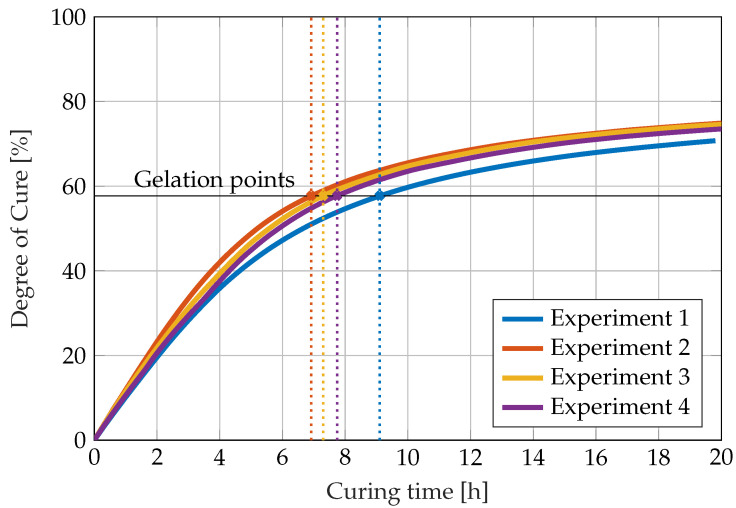
Degree of cure calculated from measured temperature data during the curing reaction of RIMR 426 resin with RIMH 435 curing agent using a kinetic model.

**Figure 7 sensors-22-07301-f007:**
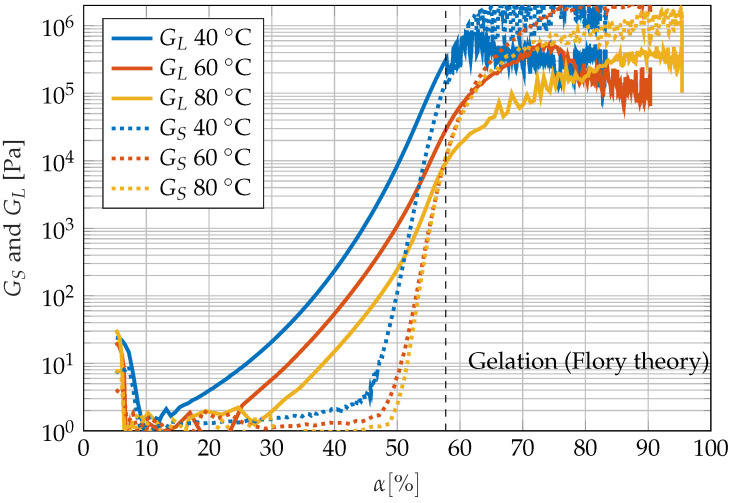
Comparison between gelation from rheometer test and gel point from Flory theory. $G_{S}$ and $G_{L}$ are storage and loss moduli of the epoxy resin.

**Figure 8 sensors-22-07301-f008:**
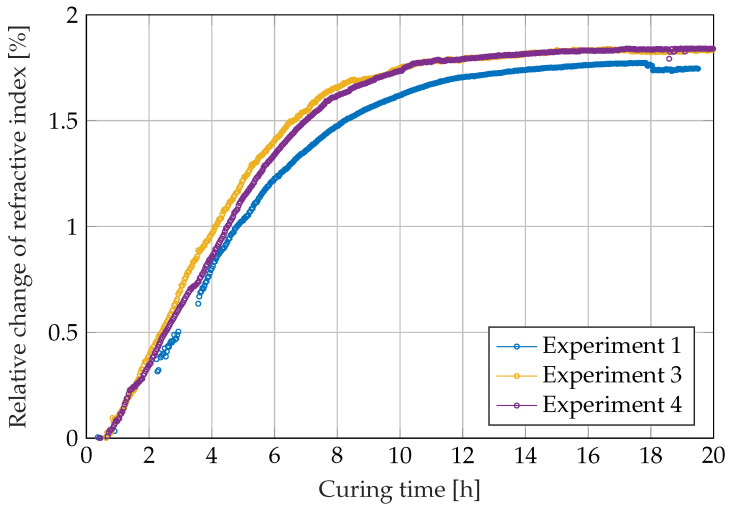
Relative changes in the refractive index during the curing reaction of RIMR 426 resin with RIMH 435 curing agent calculated from the light dot movement.

**Figure 9 sensors-22-07301-f009:**
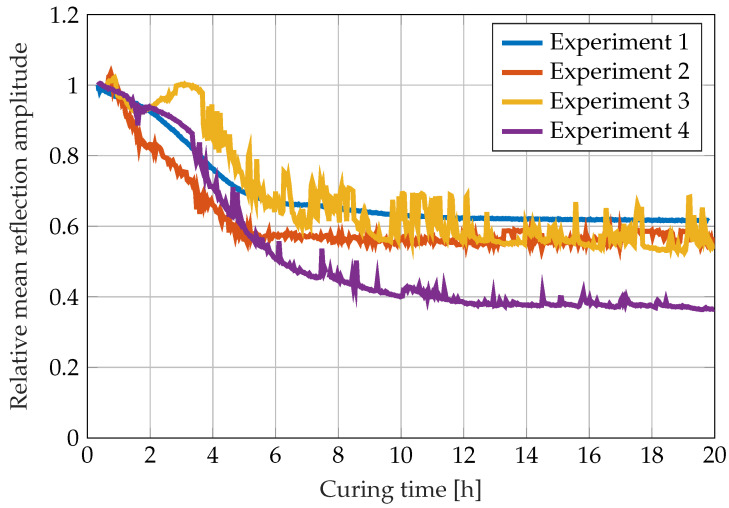
Course of the mean reflected amplitude in acoustic measurement during the curing reaction of RIMR 426 resin with RIMH 435 in relation to the initial mean value of the reflected amplitude.

**Figure 10 sensors-22-07301-f010:**
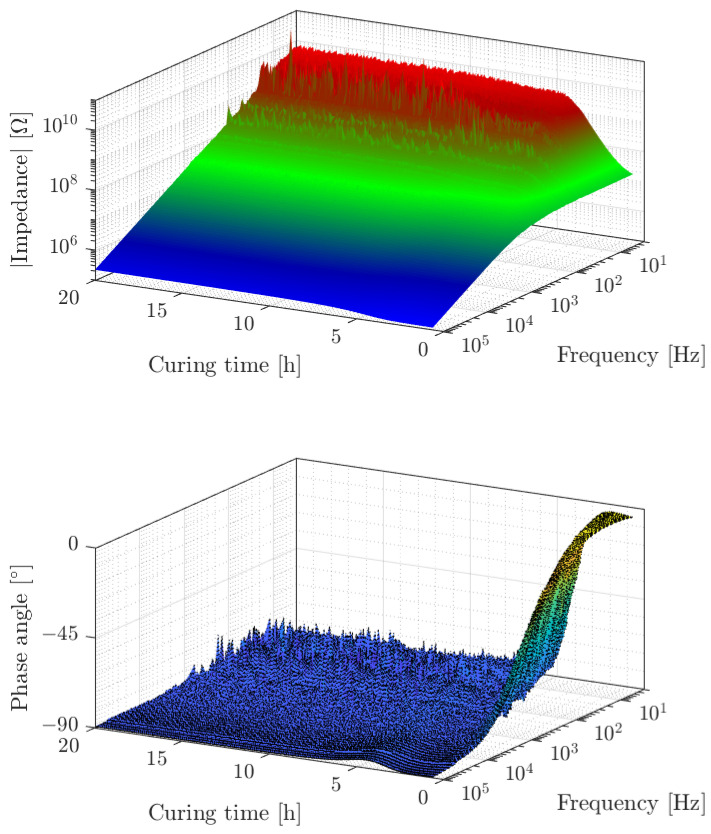
Changes in the electrical impedance of interdigitated electrodes sensors. The upper diagram shows the absolute value of the impedance, and the lower diagram shows the phase angle. The displayed data belong to experiment 2.

**Figure 11 sensors-22-07301-f011:**
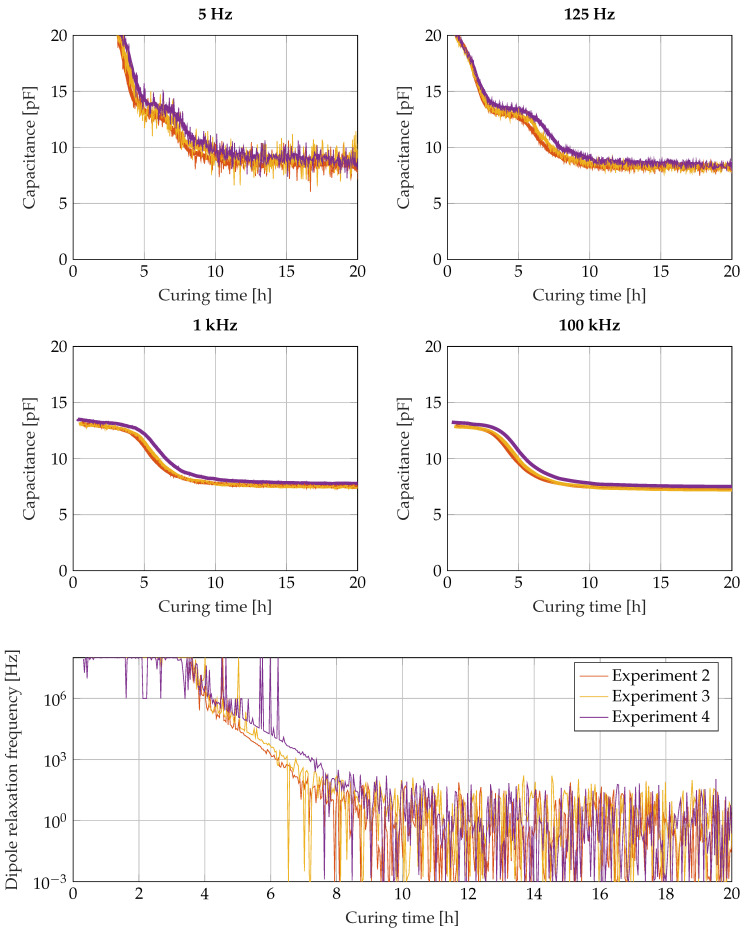
Changes in the permittivity caused by the curing reaction yield changes in the capacitance of interdigitated electrodes. The displayed capacitances were measured at 5 Hz, 125 Hz, 5 kHz and 100 kHz. In the bottom diagram, the course of the dipole relaxation frequency $f_{CC}$ is displayed.

**Figure 12 sensors-22-07301-f012:**
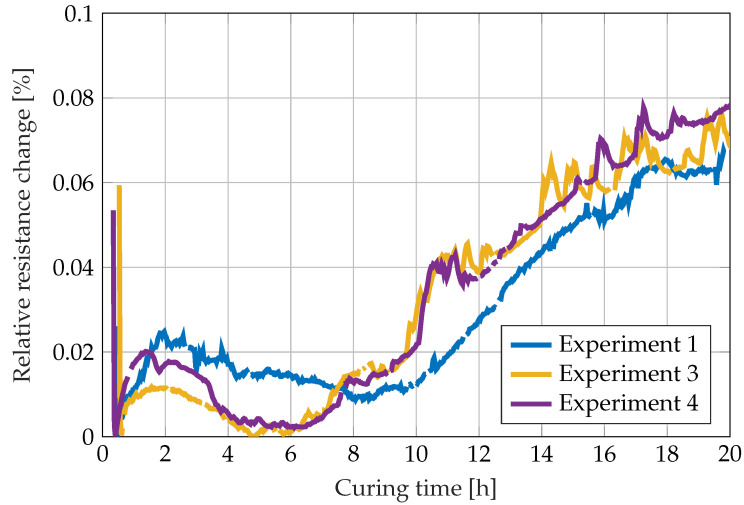
Strain development during the curing reaction of RIMR 426 resin with RIMH 435 curing agent measured by the relative resistance change of a commercial strain gauge.

**Figure 13 sensors-22-07301-f013:**
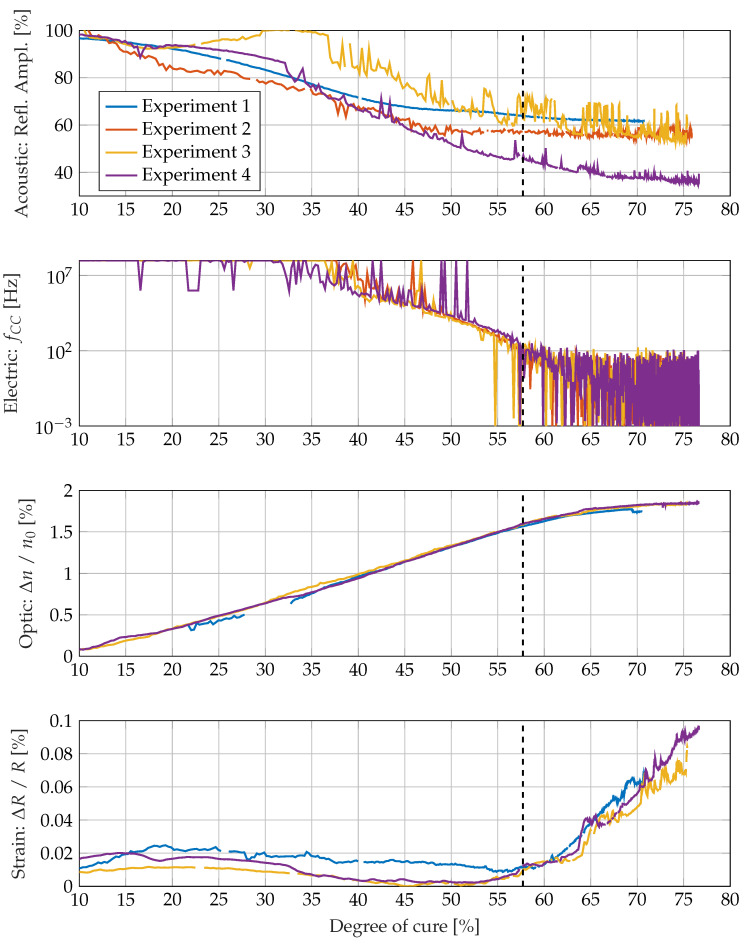
Sensitivity of different measurement parameters and derived variables for the degree of cure in different cure stages. The gelation point is shown at 57.7% degree of cure according to Flory theory.

**Table 1 sensors-22-07301-t001:** Thresholds used by the image processing algorithm for each experiment.

Experiment Number	Threshold Laser Point	Threshold Reference
1	160	30
3	180	30
4	180	30

**Table 2 sensors-22-07301-t002:** Values of fitting parameters found for Equation (Equation 4) to describe the curing behaviour of RIMR 426 resin with RIMH 435 curing agent.

$A_{1}$	$7527.8\;s^{-1}$
$A_{2}$	$4463.5\;s^{-1}$
$E_{1}$	47,578.9 J mol${}^{-1}$
$E_{2}$	40,843.3 J mol${}^{-1}$
*n*	2.929
*m*	2.079
*l*	1.144

**Table 3 sensors-22-07301-t003:** Components of the RIMR 426 resin with RIMH 435 curing agent.

Percentage	Component
Resin	
35–50%	Bisphenol-F
35–50%	Bis(4,4’-glycidyloxy)-propan
10–20%	1,4-Bis(2,3-epoxypropoxy)butan
Curing agent	
50–75%	3-Aminomethyl-3,5,5-trimethylcyclohexyl
20–25%	Trimethylhexan-1,6-diamin
10–20%	Polyoxypropylendiamin
10–20%	m-Phenylenbis(methylamin)

**Table 4 sensors-22-07301-t004:** Overview over the different experiments.

Experiment No.	Acoustic	Dielectric	Strain	Temperature	Refractive
1	✓	✗	✓	✓	✓
2	(✓)	✓	✗	✓	✗
3	(✓)	✓	✓	✓	✓
4	(✓)	✓	✓	✓	✓

## Data Availability

The test data generated during the experiments can be requested from the authors.

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
