# Peer review of "Comparison of Different Cure Monitoring Techniques"

_sensors, 2022, doi:10.3390/s22197301_

Round 1

Reviewer 1 Report

In this manuscript, the authors employ several different methods to monitor the curing process of epoxy resins and compare and discuss these monitoring techniques. This manuscript is substantial, data-rich, and well-structured. Reliable curing monitoring methods are of great significance to the research and application of epoxy resins. However, there are some problems in the manuscript, and the authors need to make major revisions to the manuscript.

1. What are the factors that affect the curing process? Such as temperature, humidity, light, etc.? Are these factors considered in the design and selection of monitors?

2. Why choose to compare these monitoring methods instead of other methods like Raman and Ultraviolet-visible Spectrum?

3. The innovation of the manuscript should be more prominent.

4. This manuscript compares several monitoring techniques. There should be some criteria for evaluating the merits of a detector. I suggest that it is better to add a unified evaluation criterion in the manuscript, such as the accuracy of the detector.

5. It is recommended to put the data together for comparison, for example, the data of several monitors can be combined into one picture. This makes it easier for readers to read.

6. It would be better if the data in the manuscript were presented more concisely and clearly. Some more important data can be extracted, such as the relationship between gelation point, time to complete curing, corresponding capacitance, and frequency in Figure 11. can be extracted to a figure.

7. For monitoring the curing process of the same sample, the values of gel point and curing time monitored by these different monitoring methods should be longitudinally compared.

Reviewer 2 Report

In the reviewed paper an important practical problem of cure state monitoring with different approaches relying on various physical phenomena is considered. While the paper content is definitely within the scope of Sensors journal, the obtained results might be of an emerging interest for a broad range of applications related to the manufacturing of composite materials.

Nevertheless, some slight improvements could be suggested, which might allow increasing the paper soundness:

1) It might be decided that the main objective of the paper is to evaluate various curing monitoring strategies suitable for their further usage in composite manufacturing. With this respect, the authors are suggested to discuss in a bit more detailed way to what extent the considered experimental setup (and specimen) could be related to the manufacturing process of a typical composite structure for, e.g., aeronautic applications? The authors might also provide their insight regarding the applicability of the proposed monitoring techniques in real-life applications (in particular, optical method)

2) In the Introduction, it might be suggested to mention at least slightly some recent advancements regarding the applicability of elastic guided waves (e.g., papers https://doi.org/10.1002/pc.25811, https://doi.org/10.1016/j.ymssp.2020.107397, etc) and modern fiber optic sensors (e.g., https://doi.org/10.1016/j.compstruct.2020.112861) for curing monitoring.

3) In subsections 2.1 - 2.3 the peculiarities of the experimental approaches are described, while the final photo is provided sufficiently further on related the paper text. It might be suggested, for clarity, to place Figure 4 in the beginning of the second section.

4) Regarding the recommendations from lines 288-291, it might be suggested not only to provide them but also carry out corresponding experiments and include these results to the paper.

5) What kind of glue was employed to attach piezoactuators to the 3mm aluminium plate?

Round 2

Reviewer 1 Report

In their revised version the authors bave addressed all the points raised in my report. The paper can be accepted in present form.